# Violence Against Nurses: Personal and Institutional Coping Strategies—A Scoping Review

**DOI:** 10.3390/bs15091166

**Published:** 2025-08-27

**Authors:** Greys González-González, Darling Rebolledo-Ríos, Ximena Osorio-Spuler, Nancy Rudner, Constanza Peña-Barra

**Affiliations:** 1PhD Program Methodology of Biomedical Research and Public Health, Universidad Autonoma de Barcelona, 08193 Barcelona, Spain; greys.gonzalez@ufrontera.cl; 2Nursing Department, Universidad de La Frontera, Temuco 4780000, Chile; 3Hospital Dr. Hernán Henríquez Aravena, Temuco 4780000, Chile; d.rebolledo05@ufromail.cl; 4Center of Excellence for Physics and Health Engineering (CFIS), Universidad de La Frontera, Temuco 4780000, Chile; 5Independent Researcher, Casselberry, FL 32707, USA; nancy.rudner@gmail.com

**Keywords:** nursing, workplace violence, exposure to violence [MeSH], mitigation strategies

## Abstract

Violence against nurses in the workplace is a worldwide concern. The high prevalence of these events has negative impacts on professionals, including stress, abandonment of the workplace, and post-traumatic stress syndrome. It is a frequent problem for nurses. As awareness of this problem increases, strategies for prevention and management of aggression and violence have evolved. This study aims to identify strategies, both institutional and personal, to address violence against nurses in the workplace. Methods: A scoping review was conducted with the PRISMA approach, using New Rayyan platform and CEMB for the evaluation of methodological quality. We included all research that studied strategies against workplace violence for nurses in hospitals in Spanish or English published between 2019 and 2024. Results: Among the 28 analyzed full-text studies, two central categories emerged with respect to addressing violence against nurses before (prevention), during (mitigation), and after (response) such events: (1) training and nurses’ action strategies and (2) practical implementation tools. Institutional leadership supporting a zero-tolerance culture; training and resources for early identification of risks; and mitigation strategies with strong emphasis on de-escalation of potential violence, building personal resilience, and support from security personnel are among the effective strategies. Conclusions: Strategies for preventing and handling workplace violence are multidimensional. Leadership engagement, addressing gender biases, conflict management training, resilience building, and security can reduce violence against nurses and its sequelae. It is essential to generate practical knowledge that is easy to apply in healthcare settings. More research is needed, especially in Latin America.

## 1. Introduction

Workplace violence and harassment have been defined as “any act or threat of physical violence, harassment, intimidation, and other threatening and disruptive behavior that occurs in the workplace with the intent to abuse or injure the victim” ([2]). The World Health Organization (WHO) points out this affects all groups of healthcare workers and all work environments ([55]). Globally, 62% of health workers have suffered violence in the workplace. The most common forms are verbal aggression (58%), followed by threats (33%) and sexual harassment (12%) ([33]). Healthcare workers in Latin America have high levels of exposure to violence, with a prevalence of 63%, only behind Asia (65%) and Australia (71%), with nurses experiencing the highest amount of mistreatment (59%) ([47]). While studies in Europe indicate that the prevalence of bullying in nurses varies between 9% and 74% ([48]), in the United States, reported rates have ranged from 21.3% to 82%. These rates demonstrate that violence against nurses is a very common and widespread phenomenon that is so normalized that is considered part of clinical practice by nurses ([13]; [41]). In a study on characteristics of the perpetrators of violence, patients’ family members were found to commit 58.6% of acts of violence, while patients committed 38.2%. The prevalence of attacks on nurses gradually increased between 2013 and 2016 ([56]). Aggressions peaked during the pandemic, when healthcare professionals, especially nurses, experienced twice as much violence—both physical and verbal ([59]). The greatest proportions of these events were reported in emergency services, pediatrics, and gynecology and were predominantly against female nurses ([40]). Although the rates declined somewhat after the peak of the pandemic, the prevalence of violence against nurses remains unacceptably high.

The prevalence of violence against female nurses ([40]) is especially worrisome in such a predominantly female profession. This pattern reflects gender-based violence, a broad societal problem, as reflected in World Health Organization Sustainable Development Goal 5, which calls on governments to address gender-based violence ([53]).

The consequences of violence in the workplace are classified as physical, professional, and organizational. Violence makes professionals suffer, causing stress, sleeping difficulties, post-traumatic stress disorder, and abandonment of the work environment ([13]; [59]). These effects are also considered detrimental to the proper functioning of a healthcare organization, with increased turnover rates and low employee retention contributing to short staffing and economic impacts ([2]; [47]).

Studies highlight how the health and safety of nurses should be a priority in the agenda of health administrators ([40]). Evidence-based initiatives to measure, prevent, control, and curb violence in the workplace are needed to ensure a violence-free work environment ([44]). Therefore, it is relevant to conduct research that investigates personal and institutional strategies for prevention and mitigation, as well as assistance for nurses who have suffered violence in their work environments. Violence against nurses does not only occur against nurses as individuals but also stems from the wider organizational context and systemic factors ([25]).

Some strategies that have been proposed to address violence in work settings are coping strategies ([20]; [60]). These can be implemented directly by nurses, based on their experience, or emerge from institutions through regulations and reforms. It is suggested that such strategies can mitigate the mental health and work performance impacts of workplace violence ([25]).

Even though workplace violence against healthcare personnel has been studied widely, a review of updated evidence is constantly needed. Healthcare managers need updated evidence on individual and institutional coping strategies that help to manage violence in health care. Because nurses are essential to health care and are often the victims of workplace violence, the present study aims to identify coping strategies, both personal and institutional, to address violence nurses experience in the workplace.

This work is framed as a first approach of the research team, prior to a project of practical implementation of these strategies in work environments in Temuco, Chile.

## 2. Materials and Methods

The present research was organized as a scoping review, guided by the specific guidelines and protocols of the Preferred Reporting Items for Systematic Reviews and Meta-Analyses (PRISMA) ([44]). The Rayyan^®^ active learning tool ([54]) was used for item selection. The quality of the articles was assessed using the critical appraisal checklist developed for survey studies by the Center for Evidence-Based Management (CEBM) ([11]).

### 2.1. Source of Information and Search Strategy

An exhaustive search of the existing literature was performed in three independent databases: Medline through PubMed, Web of Science, and Scopus. The search was carried out between August and September 2024. Descriptors were identified utilizing Medical Subject Headings (MeSH) with the terms “Workplace Violence”, “Health workers”, and “Nursing”, using free terms as synonyms linked by the Boolean operator “OR” and between “AND” concepts (Table 1).

### 2.2. Eligibility Criteria

We included all research that studied coping strategies against workplace violence in nurses published between 2019 and 2024. The target population was nurses working in any hospital setting, as well as nurses who have suffered violence from patients or family members. We included studies written in Spanish or English, without restriction in terms of research design. Studies that investigated violence from peers and other health workers, systematic and scoping reviews, case reports, theses, abstracts of conference proceedings, and opinion articles were excluded from this study. Articles investigating strategies outside the hospital setting were also excluded.

### 2.3. Original Selection of Studies

The search was performed independently and simultaneously by two researchers in the three databases. The searches were then downloaded and uploaded to the Rayyan application, which automatically eliminated duplicates. Two researchers compared the titles and abstracts to the inclusion and exclusion criteria. Only studies that contained the keywords in the titles and abstracts were included. Then, a research assistant obtained the complete texts, which were uploaded to Rayyan to be accessed by two researchers independent of each other. In the case of disagreement in the selection, a third researcher reviewed the articles and the discrepancies, reaching a degree of agreement of 1 on the Kappa index. The reasons for exclusion of texts were specified, labeled in the software as ‘Wrong design’, ‘Wrong outcome’, ‘Wrong population’, or ‘Foreign language’. The first author rechecked the final list of included studies against the inclusion and exclusion criteria to ensure consistency and accuracy.

### 2.4. Data Extraction

The data were extracted by the fourth co-author from the full-text versions of the selected articles, which were uploaded to the Rayyan platform. The following data were extracted to present a table of the main characteristics of the studies: authors, country, year, objective, methodology—sample, type of intervention, and proposed strategy.

### 2.5. Quality Assessment Methodology

The quality of evidence was assessed using the CEBM checklist ([11]). For cross-sectional and controlled studies, we used a checklist with 12 criteria, and for qualitative studies, we used a checklist with 10 criteria. Studies meeting half of the criteria were considered to be of satisfactory quality, and those meeting 75% of criteria were considered to be of good quality.

### 2.6. Analysis and Synthesis of Results

A narrative synthesis approach was used to analyze and summarize the findings of the included studies. Due to the variety of designs and models proposed, it was not possible to perform a meta-analysis. The results were categorized and summarized according to the thematic factors related to coping strategies in situations of violence against nurses that occur within hospitals. This involved a complete reading of the articles, identifying and categorizing the factors evidenced as coping strategies. The narrative approach allows for an exhaustive presentation of the strategies applied at both the personal and organizational levels among nurses who have suffered some type of violence in their work environments, making it possible to establish guidelines for future behaviors.

## 3. Results

### 3.1. Selection of Included Studies

The initial search yielded a total of 1576 articles, which were entered into the Rayyan program from the databases, automatically detecting DOI and titles, with a similarity index of 94% and a total of 683 duplicates, leaving a total of 893 articles for reading of titles and abstracts. After this process, 795 studies were excluded, leaving 98 articles for full-text review. Among these 98, 9 articles were unretrievable. Additional articles were excluded: 32 that were not congruent with the objective of this study, 5 in a language other than Spanish or English, 5 that did not focus on nurses, and 4 with an incorrect design, leaving 36 articles that fully met the inclusion criteria. Figure 1 shows the literature selection process, using the PRISMA diagram.

### 3.2. Characteristics of the Studies

Regarding the country of origin of the authors of the studies, the countries with the highest density are China, the United States, Taiwan, Australia, Brazil, India, South Africa, and Iran, in decreasing order (6, 6, 4, 4, 3, 2, 2, and 2, respectively). Other countries, such as Italy, Canada, Brunei Darussalam, New Zealand, Turkey, Egypt, and Finland, presented one study each. Only three of the studies (all Brazilian), were from Latin America. No studies emerged from Spanish Latin America. this distribution is presented in Figure 2.

In relation to the years of publication, there is no significant variation, with articles arising from 2019 to 2024.

The topics covered were grouped into the following categories: action strategies perceived by nurses, with 21 studies, and practical implementation strategies, with 15 studies. The following main characteristics were systematized: authors, country, year, objective, methodology—sample, and type of intervention, as presented in Table 2.

### 3.3. Assessment of Methodological Quality

The methodological quality of the studies included in this exploratory review is high; all met more than 75% of the criteria in the CEBM evaluation, with 31 articles with “good quality” and 5 articles with “satisfactory quality”; no article was rejected due to its methodological quality. Articles that worked with mixed methods were evaluated with both checklists according to the methodology. The detailed evaluation is presented in Table 2.

This evaluation using the CEBM criteria minimizes the risk of bias and provides a solid basis for assessing the quality of the articles and the presented information.

### 3.4. Action Strategies Perceived by Nurses

Nurses’ perceived strategies for managing violence in work settings can be grouped into three categories that respond to specific moments: before, during, and after the event of violence. In turn, nurses perceive these moments to be influenced by institutional strategies and policies and by personal and peer support strategies, which are described below.

#### 3.4.1. Before the Event of Workplace Violence

Nurses identify that training, and early interventions are essential to prevent episodes of violence ([46]). [35] ([35]) highlight training in specific skills, such as restraint and self-defense techniques, as well as education on violence triggers. [8] ([8]), in their research, also highlight the importance of training nurses in defense techniques, staff well-being, and mental health interventions.

[15] ([15]) highlight training in effective communication and the promotion of cultural exchange. Similar results were obtained by [16] ([16]) in their study of emergency healthcare workers. Their interviews with physicians and nurses in emergency highlighted both aspects (communication and education) as fundamental for the prevention of violence. Cultural change is also supported by the study conducted by [27] ([27]). Within this subcategory, [10] ([10]), highlight the implementation of security guards, controls on access to facilities, and regulation of visits as a preventive, early detection and prompt response strategy. This is also supported by [46] ([46]), who points out that increasing the number of security personnel is one of the best recommendations to control violence against workers, and [42] ([42]), who stress the importance of requesting guards immediately upon detection of an emergency situation.

#### 3.4.2. During the Event of Workplace Violence

Nurses perceive the need for concrete strategies to manage situations and protect their integrity. [14] ([14]) propose patient-centered situational management and the use of tolerance when the condition of the aggressor patient is understood, in addition to police assistance. [51] ([51]) adds the encouragement of others to act during the violent situation. On the other hand, the study by [49] ([49]) establishes guiding victims to safe areas, facilitating communication with hospital administration, and peer support as practical strategies.

#### 3.4.3. After the Event of Violence in the Workplace

Post-incident support focuses on the emotional and psychological recovery of nurses. [27] ([27]) and [35] ([35]) recommend team meetings to analyze the events; support networks among colleagues; and the use of adaptive strategies such as humor, resilience, and finding time to recover. In turn, [3] ([3]) emphasize the need for immediate psychological support and referrals to assistance services for workers who have suffered violence.

The study by [42] ([42]) highlights that among the strategies most used by nurses are informing the nurse in charge and filing lawsuits if exposed to physical violence. This could be related to the findings of [32] ([32]), who highlights, based on nurses’ experiences, the strategy of “Talk, Report, and Accept”.

#### 3.4.4. Organizational and Institutional Strategies Against Workplace Violence

Institutional policies and organizational support play a fundamental role in the prevention and management of workplace violence. According to [1] ([1]), [15] ([15]), and [8] ([8]), nurses perceive as necessary the implementation of continuous training programs, the creation of violence prevention and monitoring units, the establishment of codes of ethics, standardized procedures, security presence, and duress alarms. In addition, the implementation of “zero tolerance” policies towards violence and the increase in security personnel are identified as helpful strategy in studies by [39] ([39]) and [20] ([20]). [22] ([22]) and [50] ([50]) found that nurses perceived additional measures could be helpful if provided by the organization. These include improvements in infrastructure and adequate staffing, as well as the creation of a safe environment that considers the mental health and well-being of the staff. Finally, [39] ([39]) and [3] ([3]) emphasize the importance of effective leadership by managers, who must be able to implement these policies and provide direct support to workers.

#### 3.4.5. Personal and Peer Support Strategies

Nurses highlight the need for personal and collective strategies to cope with episodes of violence. These include seeking support from family and peer networks, engaging in recreational activities, and employing mechanisms such as coping and self-regulation, according to [21] ([21]), [27] ([27]), and [4] ([4]). Personal empowerment, through self-care and resilience, is perceived as key to recovery ([3]). [24] ([24]) highlight experience as a key factor in coping with high-risk situations.

### 3.5. Applied Implementation Strategies

This category reveals a diverse and multifaceted picture of effective interventions to address violence in the healthcare workplace. These interventions can be categorized into three main areas of practical implementation: simulation and training programs, technological intervention tools, and structured prevention and response models. Each of these categories brings distinctive and complementary elements to the comprehensive management of workplace violence.

#### 3.5.1. Simulation and Training

Six articles showed that simulations and training, especially when specifically designed for learning about situations of violence and its management, yielded significant improvements in the self-perception and confidence of nursing personnel. [37] ([37]) emphasize the importance of the development of specific and structured simulations to train professionals in situations of workplace violence, allowing for practice in a safe and controlled environment. Such simulations are particularly effective in building confidence and developing practical skills in response to stressful situations. This is supported in research by [43] ([43]), which delves into simulation as a teaching strategy for conflict de-escalation as a technique used in situations of risk of violence, focusing mainly on decreasing tense situations. The authors established a comprehensive set of fundamental principles that include maintaining calm and control in crisis situations, actively validating the feelings of both patients and family members, maintaining an appropriate physical distance during potentially violent encounters, and providing clear and continuous explanations about the procedures and situations under development. A highlight of this approach is the implementation of 15 min post-incident feedback sessions, which allow for immediate evaluation and collective learning with respect to the team’s response to crisis situations. In their study, [57] ([57]) found 104 h of de-escalation training completed in three months based on the five-component scaling technique to be helpful. The five components are communication, response, solution, care, and evaluation of the environment. Similarly, [17] ([17]) reports positive responses of nurses to an intervention involving a 4 h behavior management training course designed specifically for nurses. The course included educational materials on communication skills, verbal de-escalation techniques, and understanding the stages of violent behavior, which are crucial for identifying potential escalation and mitigating risks.

[12] ([12]) implemented a 12-session simulation training on the identification of high-risk patients, communication skills, and conflict management. This improved participants’ commitment to violence prevention and confidence in the management of violence.

Research by [31] ([31]) on the implementation of a comprehensive training program called CARE, which seeks to develop and improve resilience in nurses, including case studies, group discussions, and exchange of experiences, showed positive results among professionals.

#### 3.5.2. Technological Intervention Tools

Two studies stand out for their use of technological interventions to reduce stress. [28] ([28]) developed and evaluated two innovative therapeutic modalities: traditional Biofeedback Training (BT) and Biofeedback Training using mobile devices (SDBT). Traditional BT incorporates several techniques, including self-guided muscle relaxation, diaphragmatic breathing, rhythm-controlled breathing, and real-time respiratory sinus arrhythmia biofeedback. The SDBT modality, on the other hand, represents a technological evolution that allows professionals to practice these techniques outside the stressful work environment, offering flexibility in the implementation of stress management strategies, which do not necessarily need to be performed in the work environment. Both interventions have demonstrated significant positive effects in terms of building resilience and reducing occupational stress, with a particular advantage of SDBT in terms of accessibility and adaptability to the individual needs of healthcare personnel.

Within the line of technological tools, [59] ([59]) tested a digital platform for education and training based focused on violence in the workplace, demonstrating reductions in the incidence and severity of violence in the workplace and improving the coping resources of nurses. [7] ([7]) point out how proactive identification of risks through team meetings and alert systems allows for preventive action, in addition to the application of technologies, such as electronic signaling systems and panic buttons, which allow security personnel to be alerted to potentially violent situations and provide timely support.

#### 3.5.3. Structured Models of Prevention and Response

Six articles addressed models easily adaptable to healthcare settings to prevent violence in the workplace. [9] ([9]) present a systematic and comprehensive approach that includes detailed flow charts for the prevention of incidents according to individual characteristics of stressful situations and specific management principles and effective communication strategies. This achieved significant reductions in the incidence of psychological and physical violence against nursing staff. This structured approach is enriched and complemented by the coping model developed by [36] ([36]), who establish a comprehensive framework that encompasses multiple dimensions, including the systematic identification of fear triggers, a detailed assessment of coping needs, the implementation of specific methods, and their subsequent continuous re-evaluation to ensure their effectiveness. Among the implementation models, [26] ([26]) demonstrated the need for strategies with a zero-tolerance model of violence in which staff training, policy development, and a behavioral emergency response team trained in stress reduction techniques are implemented.

[34] ([34]) implemented a multicomponent violence reduction model called “Safewards” with 10 basic interventions based on the theory of mental health (clear mutual expectations, soft words, talk down, positive words, mitigation of bad news, getting to know each other, mutual help meeting, methods to calm down, safety, and high-communicative-efficiency messages). Nurses in the study found the two most effective strategies were “soft words” and “talk down”. These two interventions were promoted with posters in the units with increased threats of violence.

[45] ([45]) proposed a model with (1) behavioral standards, i.e., an intervention involving the establishment of unit-specific behavioral expectations to guide staff and visitors on acceptable conduct within the healthcare environment; (2) a de-escalation algorithm, i.e., a management algorithm developed to provide staff with strategies for de-escalating potentially aggressive situations; and (3) staff education, with comprehensive training provided for staff members on how to recognize and manage escalating behaviors effectively. The intervention effectively addressed workplace violence and aggression, leading to improved safety perceptions and a notable reduction in aggressive incidents.

### 3.6. Practical Implications and Recommendations

The integration of the various strategies and practical implementation tools suggests that a multimodal approach to workplace violence prevention and management is the most effective. This combination provides healthcare institutions with a comprehensive set of resources to address this complex challenge. Evidence suggests that the coordinated implementation of these different strategies can create a safer and more resilient work environment in which healthcare workers feel better prepared and supported to deal with situations of potential or actual violence. An intervention proposal, as presented in Figure 3, is suggested through a decision and characterization diagram for use by managers and decision makers in work environments threatened by violence toward nurses. The diagram arises from the analysis of the complete texts, followed by rereading and characterization, with special emphasis on the opportunity that this analysis suggests with strategies that can be added to hospital environments where nurses are victims of violence by patients and family members.

## 4. Discussion

### 4.1. Action Strategies as Perceived by Nurses

Three phases fundamental in the management of violence in the workplace emerge from the studies: before, during, and after workplace violence. Training is perceived as fundamental to prevent and manage violence, addressing the recognition of risk situations, communication, and de-escalation strategies ([27]; [22]; [35]). Multiple studies highlight poor communication among teams as a major contributor and cause of physical and non-physical violence in the workplace ([41]; [29]; [16]). Additionally, the need for participation in ongoing training programs on strategies to prevent and to manage violence is constant ([18]).

The preventive strategies highlighted in the studies include the implementation of security guards, controls on access to facilities, and regulation of visits ([46]; [42]; [8]). Notably, health professionals who think that security services respond adequately to incidents of violence are more likely to feel safe in work environments ([38]). Despite the fact that these strategies have a high level of acceptance, only 7.8% of situations of violence are handled by security agents ([38]).

During incidents of violence, effective strategies include patient-centered situational management, with tolerance and understanding of the situation ([46]), in addition to removing the victim to a safe space to protect his or her integrity, improving communication with regulatory centers, peer support, and police assistance ([39]; [10]).

Post-incident support focuses on communication to the charge nurse and emotional and psychological recovery of nurses; team meetings to analyze the events; encouraging support networks; the use of coping strategies such as humor, resilience, and finding the time to recover; and immediate psychological support ([54]; [15]; [19]; [32]).

Studies indicate that the implementation of legal aspects and policies is fundamental to addressing workplace violence ([29]; [6]). The results of the study show that nurses also attach special importance to institutional policies of support and prevention of situations of risk of violence ([35]; [1]; [42]). Other organizational measures are infrastructure safety improvements, adequate staffing, and the creation of safe environments that consider the mental health and well-being of the staff ([20]; [50]; [51]; [4]). The commitment of hospital administrators is necessary to guarantee and ensure a safer work environment for health professionals ([23]).

In addition, nurses emphasize the importance of seeking support from family members and colleagues ([21]; [3]). Peer support and support seeking have been emphasized as strategies with a positive impact on coping with the distress of being a victim of violence ([19]). Professional empowerment ([15]) and experience ([14]) are also highlighted as key factors in a better response to this type of situation. A greater number of years of experience has been related to a greater probability of reporting situations of violence ([58]). [5] ([5]) emphasize that managers should address the elements that arise from the perceptions of nurses on how to work with situations of violence, since it has been related to a positive impact on the quality of care and employee retention.

### 4.2. Practical Implementation Strategy

Simulation and training emerge as tools that demonstrate significant improvements in the self-perception and confidence of nursing staff through specifically designed programs for learning about situations of violence and their management. Simulations are particularly effective in building confidence and developing practical skills to respond to stressful situations ([37]; [43]; [31]). Identification of high-risk patients, communication skills, and conflict management re-emerge as essential topics in the management of violent situations ([57]; [12]).

These findings are supported by previous reviews, which highlight communication, for example, as an effective strategy for violence management ([18]). However, the use of validated questionnaires in which communication, de-escalation, and conflict management tools are provided has been particularly effective, and it should be noted that these trainings must have governance, quality, and review processes to ensure their success ([30]).

Regarding the second characteristic, technological intervention tools have emerged. These include cell phone and computer applications to address situations of violence, as well as electronic signaling systems and panic buttons to alert security personnel of potentially violent situations and provide timely support ([28]; [7]).

Finally, structured prevention and response models are described, which are easily adaptable to healthcare settings to prevent violence at work. Systematic approaches were presented based on the individual characteristics of stressful situations ([9]), The coping model ([36]) proposes a comprehensive framework covering multiple dimensions—from the systematic identification of fear triggers, through an assessment of coping needs, to the implementation of specific methods and their subsequent and continuous re-evaluation to ensure their effectiveness. The implementation of the zero-tolerance model ([26]) and multicomponent models ([34]; [45]) also emerged. Programs and models of practical implementation in work environments have favored safety and confidence in nurses in the face of situations of violence and aggression ([52]).

### 4.3. Limitations of the Study

The use of three databases is the main limitation of the study, limiting the scope of the review and preventing the exhaustive coverage of relevant studies. It is recommended for future research that a greater number of databases be included to achieve a wider scope. Also, a more extensive list of keywords may have produced a larger number of relevant articles. Furthermore, the linguistic limitations of only English and Spanish may have limited the inclusion of other studies.

## 5. Conclusions

Multifactorial strategies against violence and aggression in nurses’ work environments can be implemented by health managers to avoid incidents of aggression the negative impact they can have on nurses. Interventions are strongest when the purpose is clear, concise, and well-supported by leadership. Healthcare organizations have a duty to improve work environments. Figure 3 provides a diagram of intervention proposals.

Organizational leadership can strive to create a non-tolerance culture. Training in recognizing risks, de-escalation and mitigating violence can be effective. Security systems and personnel for prevention and quick response to aggression can also reduce violence. Structured prevention and response models are also effective. Multiple strategies to prevent, mitigate, and address violence, before, during, and after such occurrences have the potential to improve the workplace and support the nursing workforce. 

While not found in this scoping review, widespread cultural patterns of gender bias may contribute to violence against nurses. The World Health Organization’s Sustainable Development Goal 5 calls for gender-based violence to be addressed.

In spite of the high prevalence of violence against health workers in Latin America, no articles were identified in Spanish Latin America, and we found only three articles from the region (Brazil). While violence against nurses has been studied in many countries, there is a clear need for additional studies in Latin American countries, which have cultural and social patterns, as well as safety needs, that vary according to the context of each country.

## Figures and Tables

**Figure 1 behavsci-15-01166-f001:**
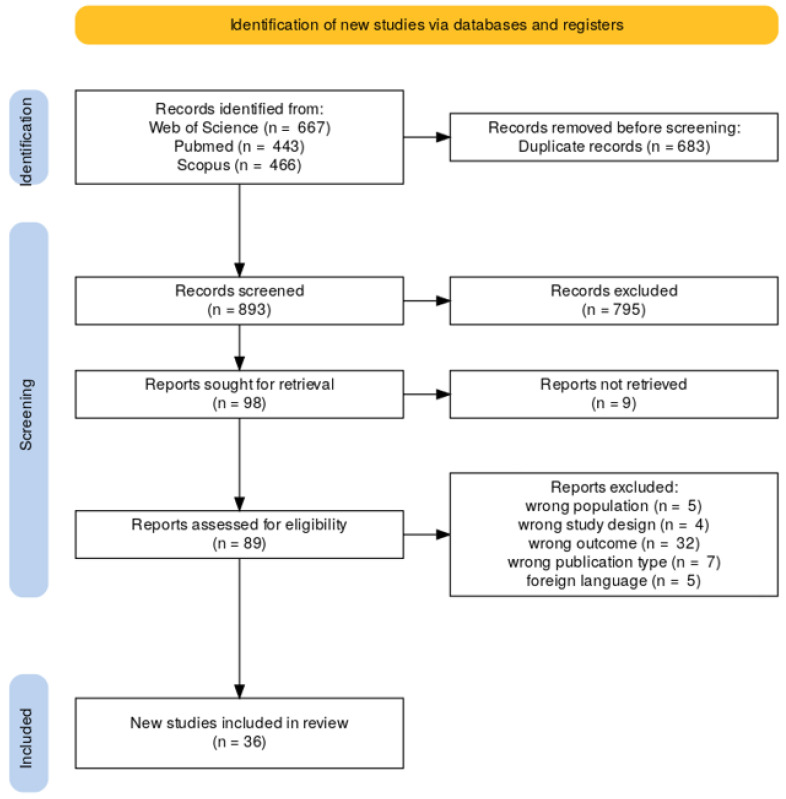
PRISMA diagram of studies identified and included in the study.

**Figure 2 behavsci-15-01166-f002:**
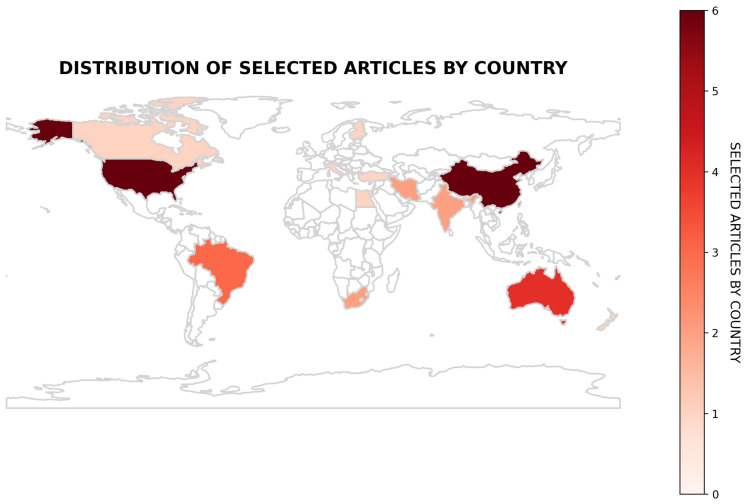
Distribution of selected articles by country.

**Figure 3 behavsci-15-01166-f003:**
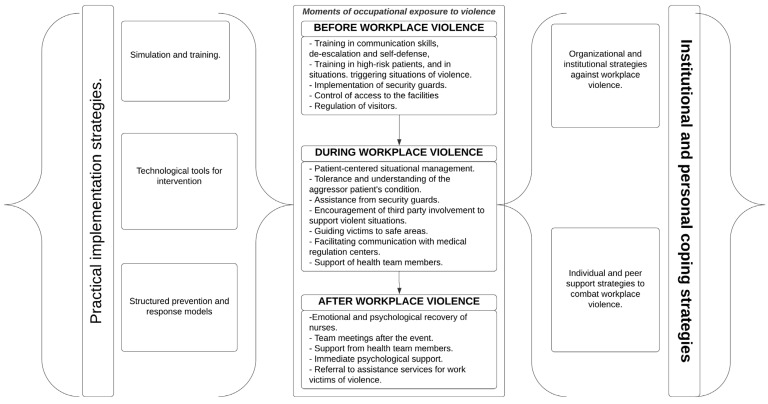
Evidence-based intervention proposal.

**Table 1 behavsci-15-01166-t001:** Descriptors used in the search string.

Workplace Violence AND	Coping Strategies AND	Nursing
Mobbing	Coping	Nurses
Bullying	Strategies	Health workers
Patient and visitor violence		
Violence against healthcare workers		

**Table 2 behavsci-15-01166-t002:** Characteristics of studies whose full texts were read.

Authors, Country, Year	Aim	Methodology	Interventions	Methodological Quality
	**Nurse-perceived strategies**			
Fan S, et al., 2021. China	To assess the impact of violence on nurses’ mental health and its related variation in resilience and coping strategies	Cross-sectional study	Questionnaire	Good quality
Hsieh H, Chen Y, Chen S, Wang H., 2023 Taiwan	Understanding the perceptions of nurses who have been victims of violence in work settings and their coping strategies	Qualitative study	Semi-structured interviews	Good quality
Han C, Chen L, Lin C, Goopy S, Lee H., 2024 Taiwan	Understanding how emergency nurses develop resilience in the context of workplace violence	Qualitative study	Semi-structured interviews	Good quality
Dadashzadeh A, Rahmani A, Hassankhani H., 2019 Ira	To explore the experiences of Iranian nurses working in pre-hospital emergencies and strategies for coping with violence	Qualitative study	Semi-structured interviews	Good quality
Rajabi F, Jahangiri M, Bagherifard F, Banaee S., 2000 Iran, 2020.	Addressing the problem of occupational violence in health care settings	Qualitative study	Interviews with guiding dialogue questions	Good quality
Gab Allah A, Elshrief H, Ageiz M., 2020 Egypt	Evaluating nursing work-related problems as perceived by managers and developing strategies for decision making	Cross-sectional study	Questionnaire, Delphi technique, and interview	Satisfactory quality
Singh A, Ranjan P, Agrawal R, Kaur T, Upadhyay A., 2023. India.	Assessing the problem of violence against healthcare workers in healthcare settings	Cross-sectional study	Questionnaire	Good quality
Davey K, et al., 2020 India	To gain a better understanding of issues surrounding violence against healthcare providers in Indian EDs	Qualitative study	Semi-structured interviews	Good quality
Lim, Z., 2023 Brunei Darussalam	Identify and explore the impact of violence on mental health nurses and discuss nurses’ coping mechanisms	Qualitative study	Electronic interview	Good quality
Öztaş İ, Yava A, Koyuncu, A., 2023 Turkey.	Determine emergency nurses’ exposure to workplace violence by patients and their relatives and the nurses’ use of coping behaviors/methods	Cross-sectional study	Semi-structured interviews	Good quality
Smith C, Palazzo S, Grubb P, Gillespie G., 2020 United State.	To explore strategies suggested by new graduate nurses to prevent and intervene during incidents of violence in work settings	Qualitative study	Interviews with guiding dialogue questions	Good quality
Myers G, Kowal C., 2023 United States	Explore encounters with verbal aggression and solutions offered by nurses	Qualitative study	Surveys and interviews	Good quality
Bekelepi N, Martin P., 2022 South Africa	To explore and describe the experiences of nurse victims of violence, as well as strategies and the support received	Qualitative study	Semi-structured interviews	Good quality
Agu A, Azuogu B, Una A, Ituma B, Eze I, Onwe F, Oka O., 2023 Africa	To uncover managerial perspectives on intervention strategies for violence in the workplace and tertiary health prevention	Qualitative study	Semi-structured interviews	Good quality
Bakes L, Mansfield Y, Meechan T., 2021 Australia	Understand staff perceptions of peer support after incidents of violence	Qualitative study	Structured interviews	Satisfactory quality
Dafny H, Beccaria G, Muller A., 2022 Australia	To determine nurses’ perceptions of management, strategies, and support services for violence in work settings	Qualitative study	Surveys with closed and open-ended questions	Good quality
Cabilan C, 2022 Australia	To explore and collate solutions for occupational violence from emergency department (ED) staff	Qualitative study	Electronic survey	Good quality
Martins, Moloney, Jacobs, y Anderson, 2023 New Zealand	Provide evidence to support nurses affected by workplace aggression and violence	Mixed methods	Lickert-type survey and semi-structured interview	Good quality
Sé A, Machado W, Gonçalves R., 2021 Brazil	Identify strategies to prevent violence in pre-hospital services	Mixed methods	Semi-structured interviews	Good quality
Carvalho K, Araujo P, Santos F, Oliveira P, Silva J, Santos K., 2023 Brazil	To analyze violence in work environments among nurses in the context of primary care	Qualitative study	Semi-structured interviews	Good quality
Florido H, Duarte S, Floresta W, Fonseca A, Broca A., 2020 Brazil	To identify situations of violence in the work routine of health professionals and describe coping strategies	Qualitative study	Semi-structured interviews	Good quality
Practical implementation strategies				
Ming, J, Huang H, Hung S, Chang C. Hsu Y., 2019 China	Effectiveness of situational simulation training with respect to nurses’ self-concept and self-confidence to cope with violence in work settings	Quasi-experimental study	Simulation and application of questionnaires in focus groups	Satisfactory quality
Ye J, et al., 2020 China	To evaluate the effectiveness of a training program on de-escalation techniques for nurses working in psychiatric units	Randomized controlled trial	Training in the application of de-escalation techniques	Good quality
Cai J, et al, 2023 China	To formulate a step-by-step approach to violence prevention strategies in work settings under the guidance of situational prevention theory and elements drawn from nurses	Quasi-experimental study	Application of strategies through classroom classes and simulation	Satisfactory quality
Zhang Y, Cai J, Qin Z, Wang H, Hu X., 2023 China	To investigate the impact of a digital platform on the incidence, severity, and coping with violence for nurses	Quasi-experimental study	Education and training platform	Good quality
Liao L, et al., 2024 China	Test the effect, feasibility, and acceptability of the Comprehensive Active Resilience Education (CARE) program in improving the resilience of emergency nurses exposed to workplace violence	Quasi-experimental study	Education and training platform	Good quality
Hsieh H, Huang I, Liu Y, Chen W, Lee Y, Hsu H., 2020 Taiwan	Compare the effectiveness of a biofeedback training phone (SDBT) and biofeedback training application among nurses who have experienced workplace violence by patients	Quasi-experimental, pre–post-test study	Tool for BT training and SDBT training.	Satisfactory quality
Chang Y. Hsu M. Ouyang W., 2022 Taiwan	To evaluate the effects of a novel integrated patient and visitor violence prevention and management training program	Quasi-experimental	12-session training	Good quality
Hendrickson S., 2022 United States	Explore one hospital’s journey to understand the implementation of mitigation strategies against violence in work settings	Implementation and follow-up strategy	Development, implementation, and training on strategies against workplace violence	Good quality
de la Fuente, M., et al., 2019 United States	Evaluate the impact of behavior management training on nurses’ confidence in managing aggressive patients	Quasi-experimental, pre–post-test study	Training in behavior management	Good quality
Poore J, Mays C, McKibban L, Harbert Z, Schroedle K., 2024 United States.	Explore the development and implementation of de-escalation training for nurses entering work settings	Simulation with qualitative and quantitative approaches to research	Conflict de-escalation simulation	Good quality
Quigley, E, et al., 2021 United States	Development of standardized strategies to enhance the professional advancement and safety of healthcare workers	Implementation and follow-up strategy	Development, implementation, and training on strategies	Good quality
Luck L, Kaczorowski K, White M, Dickens G, McDermid F., 2023 Australia	Explore the experiences of nurses in implementing and modifying “safewards” (conflict mitigation) interventions	Participatory action research	Conflict mitigation interventions	Good quality
Mikkola R, Huhtala H, Paavilainen E., 2019 Finland	To develop a model of work-related fear coping in emergency department staff with a focus on medical and nursing staff	Cross-sectional study	Survey application	Good quality
Burkoski V, Farshait N, Yoon J, Clancy P, Fernandes K, Howell M., 2019 Canada	To explore nurses’ experiences with implementing technology-based violence prevention interventions	Qualitative study	Semi-structured interviews	Good quality

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
