# Peer review of "Violence Against Nurses: Personal and Institutional Coping Strategies—A Scoping Review"

_behavsci, 2025, doi:10.3390/bs15091166_

Round 1
Reviewer 1 Report
Comments and Suggestions for Authors
Abstract
The abstract mentions violence against nurses and the need for coping strategies but does not clearly present the central objective of the study, which should be to assess which personal and institutional coping strategies are most commonly used by nurses experiencing workplace violence.
The methodological description is brief, lacking a clear explanation of the inclusion/exclusion criteria for the reviewed studies or the databases used.
The summary of the results should be improved, as it does not emphasise the key findings.
The abstract conclusion does not sufficiently highlight the practical implications of the findings for professional practice and hospital management.
Introduction
A conceptual definition of "Violence Against Nurses" is provided, but there is no conceptual definition of "coping strategies". What coping strategies exist? What do they entail? What are personal and institutional coping strategies?
The justification for this study is not clearly addressed. Why is this research important?
The study’s objective is unclear. What does it aim to assess? For example, evaluating which personal and institutional coping strategies are used by nurses who experience workplace violence.
Methodology
One of the descriptors used twice is "Health workers". Why? The study focuses on nurses. Justify the repetition of this descriptor.
Although inclusion and exclusion criteria are mentioned, some key details are missing, such as the target population. Does it include all nurses across all care settings? Does it focus specifically on studies addressing coping strategies?
Results
The results do not clearly present the personal and institutional coping strategies used by nurses. It is suggested that a table be created to summarise the coping strategies. Which coping strategies were most commonly used, considering the analysed studies?
Discussion
Figure 3 refers to some of the key findings on coping strategies, but there is no prior table emphasising which studies these findings were derived from. This table would be better placed in the Results section.
The discussion does not sufficiently emphasise how the analysed strategies can be implemented in the hospital setting.
Conclusion
The conclusion does not clearly summarise how the results address the research objectives.
More concrete proposals could be made to further investigate workplace violence against nurses.
Specific guidelines relevant to this study for hospital managers and healthcare professionals could be included.
Author Response
Comentario 1: El resumen menciona la violencia contra el personal de enfermería y la necesidad de estrategias de afrontamiento, pero no presenta con claridad el objetivo principal del estudio, que debería ser evaluar qué estrategias de afrontamiento, tanto personales como institucionales, utilizan con mayor frecuencia el personal de enfermería que experimenta violencia laboral.
La descripción metodológica es breve y carece de una explicación clara de los criterios de inclusión/exclusión de los estudios revisados o de las bases de datos utilizadas.
El resumen de los resultados debería mejorarse, ya que no enfatizar los hallazgos clave.
La conclusión del resumen no destaca suficientemente las implicaciones prácticas de los hallazgos para la práctica profesional y la gestión hospitalaria.
Respuesta 1: Se modificó el objetivo central del texto para aclararlo. Se revisó el comentario y se agregaron sugerencias a la sección.
Comentarios 2: Se proporciona una definición conceptual de "Violencia contra Enfermeras" , pero no existe una definición conceptual de "estrategias de afrontamiento" . ¿Qué estrategias de afrontamiento existen? ¿Qué implican? ¿Cuáles son las estrategias de afrontamiento personal e institucionales? La justificación de este estudio no se aborda con claridad. ¿Por qué es importante esta investigación? El objetivo del estudio no está claro. ¿Qué pretendo evaluar? Por ejemplo, evalúe qué estrategias de afrontamiento personal e institucional utilizan las enfermeras que experimentan violencia en el lugar de trabajo.
Respuesta 2: Con base en los comentarios del revisor, se definieron las estrategias de afrontamiento y se modificó el objetivo. Es importante aclarar que el estudio es una revisión exploratoria para determinar el alcance de la literatura disponible que pueda informarnos sobre las estrategias implementadas para abordar la violencia laboral contra el personal de enfermería. El estudio afirma que esta revisión exploratoria guiará un proyecto de implementación práctica en Temuco, Chile.
Comentario 3: Uno de los descriptores utilizados dos veces es "Trabajadores de la salud" . ¿Por qué? El estudio se centra en el personal de enfermería. Justifique la repetición de este descriptor.
Si bien se mencionan los criterios de inclusión y exclusión, faltan algunos detalles clave, como la población objetivo. ¿Incluye a todo el personal de enfermería de todos los centros de atención? ¿Se centra específicamente en estudios que abordan estrategias de afrontamiento?
Respuesta 3: Analizamos estudios sobre "trabajadores de la salud" porque estos estudios a menudo incluían enfermeras. Revisamos los artículos e incluimos solo aquellos que incluían enfermeras en los grupos de trabajadores de la salud para evitar omitir datos que pudieran ser relevantes. Aclaramos que la población objetivo incluye enfermeras dentro y fuera del ámbito hospitalario.
Comentario 4: Los resultados no presentan con claridad las estrategias de afrontamiento personal e institucionales utilizadas por el personal de enfermería. Se sugiere crear una tabla para retomar las estrategias de afrontamiento. ¿Cuáles fueron las más utilizadas, considerando los estudios analizados?
Respuesta 4: En relación con este punto, se ha añadido a esta sección la Figura 3 para mayor claridad.
Comentarios 5: La Figura 3 se refiere a algunos de los hallazgos clave sobre estrategias de afrontamiento, pero no existe una tabla previa que destaque de estudios qué se derivaron estos hallazgos. Esta tabla se ubicaría mejor en la sección de Resultados .
La discusión no enfatiza suficientemente cómo se pueden implementar las estrategias analizadas en el entorno hospitalario.
Respuesta 5: En relación con este punto, se ha añadido a esta sección la Figura 3 para mayor claridad.
Comentarios 6: La conclusión no resume claramente cómo los resultados abordan los objetivos de la investigación.
Se podrían formular propuestas más concretas para investigar más a fondo la violencia laboral contra el personal de enfermería.
Se podrían incluir directrices específicas relevantes para este estudio dirigido a gerentes de hospitales y profesionales sanitarios.
Respuesta 6: Se ha revisado la conclusión para incluir hallazgos y recomendaciones más concretas.

Reviewer 2 Report
Comments and Suggestions for Authors
I have done some work in this area and like this article a lot. There is not enough attention to this subject. The discussion section (Section 4) is particularly noteworthy for its detail and clarity. A colleague and I did a few small polls and found similar interest in our respondents in the interest in organization policies (see line 337).
There are several items that should be cleaned up before publication:
(1) The conclusions are weak and too general. There are no clear recommendations or actions as a result of this work. I am surprised by this given the quality of the discussion.
(2) The understanding of violence and the toxicity of the work environment seems narrow. The word choices in 2.1 are mostly general and mention only one specific type of workplace violence ("mobbing") and ignore other terms and conditions. Some examples: Why not the word bullying? What about the physical violence of emergency rooms and when dealing with mentally ill patients? What about the threats of violence against health workers stemming from COVID? And so on. I think the search string should be expanded.
(3) The statement of the problem in the Introduction seems to overlook the US and Europe.
(4) While the article is in English, several of the country names in column 1 of Table 1 are in Spanish, e.g., "Estados Unidos," "Nueva Zelanda." A smart person should be able to figure this out, but the use of Spanish here is inconsistent with the rest of the article.
Author Response
- Comments 1: he conclusions are weak and too general. There are no clear recommendations or actions as a result of this work. I am surprised by this given the quality of the discussion.
- Response 1: The commentary was revised and suggestions were added to the section.
- Comments 2:The understanding of violence and the toxicity of the work environment seems narrow. The word choices in 2.1 are mostly general and mention only one specific type of workplace violence ("mobbing") and ignore other terms and conditions. Some examples: Why not the word bullying? What about the physical violence of emergency rooms and when dealing with mentally ill patients? What about the threats of violence against health workers stemming from COVID? And so on. I think the search string should be expanded.
- Response 2: Dear Reviewer, we especially appreciate this comment because as a team we had left out some very relevant words such as bullying. We did the search again adding also the word “Patient and visitor violence” and “violence against healthcare workers”. We again searched in the databases Web of Science and Pubmed, and Scopus.
- Comments 3: The statement of the problem in the Introduction seems to overlook the US and Europe.
- Response 3: Data from the United States and Europe were added. Thank you for the suggestion.
- Comments 4:While the article is in English, several of the country names in column 1 of Table 1 are in Spanish, e.g., "Estados Unidos," "Nueva Zelanda." A smart person should be able to figure this out, but the use of Spanish here is inconsistent with the rest of the article.
- Response 4: Thank you for catching that. This has been corrected.

Round 2
Reviewer 1 Report
Comments and Suggestions for Authors
It does not mention how the quality assessment of the included studies was conducted: Although the CEBM checklist was used, the article does not explain how the criteria were applied or whether any studies were excluded due to low methodological quality.
The references do not comply with the journal's guidelines. It is recommended to review and correct them accordingly.
Author Response
Comments 1: It does not mention how the quality assessment of the included studies was conducted: Although the CEBM checklist was used, the article does not explain how the criteria were applied or whether any studies were excluded due to low methodological quality.
Response 1: Dear reviewer, we are especially grateful for this comment, we have made a new and more detailed evaluation, and we believe that this will also contribute to the quality of our study, in table 2, we have added a column with quality, so that readers of our article will have clarity of the quality evaluation.
Comments 2: The references do not comply with the journal's guidelines. It is recommended to review and correct them accordingly.
Response 2: Dear reviewer, thank you very much for your advice, we have reviewed and been very careful about the references, and we have followed the template provided by the journal, we have also compared with other publications of the journal to have more certainty about our references, we improved what was necessary and modified what was suggested, we remain attentive to any new suggestion.
We would also like to add that we have again checked the English of the publication, which has been done by one of our team members who is a native English speaker.
Reviewer 2 Report
Comments and Suggestions for Authors
Thank you for the corrections. Amazing how some small changes can substantively improve the study.
Author Response
Comments 1: Thank you for the corrections. Amazing how some small changes can substantively improve the study.
Response 1: Dear reviewer, we are really grateful for your suggestions, they were not only very kind, but also served us deeply to improve our study and research, we are currently working on a research project where we will apply these results to nurses in an ER unit that presents high rates of violence in Chile. Once again, thank you very much.